# Development of a Library of Disulfide Bond-Containing Cationic Lipids for mRNA Delivery

**DOI:** 10.3390/pharmaceutics15020477

**Published:** 2023-02-01

**Authors:** Zhigao Shen, Cong Liu, Ziqian Wang, Fengfei Xie, Xingwu Liu, Lingkai Dong, Xuehua Pan, Chen Zeng, Peng George Wang

**Affiliations:** Department of Pharmacology, School of Medicine, Southern University of Science and Technology, Shenzhen 518055, China

**Keywords:** lipid nanoparticles, mRNA delivery, disulfide bond

## Abstract

Lipid nanoparticles (LNPs) are the commonly used delivery tools for messenger RNA (mRNA) therapy and play an indispensable role in the success of COVID-19 mRNA vaccines. Ionizable cationic lipids are the most important component in LNPs. Herein, we developed a series of new ionizable lipids featuring bioreducible disulfide bonds, and constructed a library of lipids derived from dimercaprol. LNPs prepared from these ionizable lipids could be stored at 4 °C for a long term and are non-toxic toward HepG2 and 293T cells. In vivo experiments demonstrated that the best C4S18A formulations, which embody linoleoyl tails, show strong firefly luciferase (Fluc) mRNA expression in the liver and spleen via intravenous (IV) injection, or at the local injection site via intramuscular injection (IM). The newly designed ionizable lipids can be potentially safe and high-efficiency nanomaterials for mRNA therapy.

## 1. Introduction

mRNA vaccines have recently gained increasing attention to treat various infectious diseases, which showed the highest efficacy against the SARS-CoV-2 pandemic [1,2,3,4]. Theoretically, protein replacement therapy [5,6], cancer immunotherapy [7,8,9] and gene editing can also be reached via mRNA therapy [10]. However, the large size, negative charge and high hydrophilicity of mRNA stop it from passing through the cell membranes. Besides, the mRNA is highly immunogenic and can be sensed by the pattern recognition receptors, such as the endosomal Toll-like receptors TLR3, TLR7 and TLR8, which reduce the mRNA stability [11,12]. Furthermore, the RNase in the serum and tissues can degrade the mRNA rapidly after in vivo administration [13,14,15]. These barriers limit the medical application of mRNA therapy, and it is in urgent need for mRNA vectors. The vectors can not only protect mRNA from nuclease degradation and increase mRNA stability in plasma, but also deliver it into the target cells and organs. Therefore, many efforts have been made to explore safe and efficient mRNA delivery systems, including viral and non-viral vectors. Despite the high efficacy in mRNA delivery, viral vectors are prone to induce harmful immune-mediated responses, unwanted incorporation and other toxic side effects [16]. Non-viral synthetic vectors avoid these risks and offer unique safety advantages over viral vectors. During the past decade, a variety of non-viral mRNA delivery vehicles have emerged, such as lipid nanoparticles (LNPs) [17], polymers [18] and inorganic nanoparticles [19].

Lipid nanoparticles are the most promising non-viral vectors, and all the clinically approved mRNA vaccines for SARS-CoV-2 utilized the LNPs’ delivery platforms. LNPs are typically comprised of four components, ionizable cationic lipid, helper lipid, phospholipid and polyethylene glycol lipid. The structure of LNPs resembles the bilayer of the cell membrane and mRNA-loaded LNPs can be internalized into cells via the endocytosis pathway. Then mRNA-loaded LNPs accumulated in the endosome and gradually fused with the endosomal membrane, which enables the mRNA to escape from the endosomal compartments and be released into the cytoplasm to initiate the translation. Of the four components in LNPs formulation, ionizable cationic lipids play the most critical role. Ionizable cationic lipids interact with mRNA payloads by electrostatic complexation and are prone to be protonated, due to the low pH value in the endosome, destabilizing the membrane of LNPs and facilitating the endosomal escape of mRNA. Disulfide bonds have been widely used as biodegradable motifs for prodrugs and fluorescent probes [20,21], but this strategy seldom appeared in the development of lipids. Xu and coworkers have designed many lipid-like compounds containing disulfide bonds, but the synthetic route is complex [22,23,24,25]. Here, we design a library of disulfide bond-containing cationic lipids and use them to prepare LNPs with DSPC, Cholesterol and DMG-PEG 2000, following standard four-component formulation, which can serve as a high-efficiency mRNA delivery platform (Figure 1).

## 2. Materials and Methods

### 2.1. Chemical Reagents

The 2,2’-dithiodipyridine, dimercaprol, sulfer alcohol, oleic acid and linoleic acid were purchased from Macklin (Shanghai, China), Bidepharm (Shanghai, China) and J&K Scientific (Beijing, China). DSPC, cholesterol and DMG-PEG2000 were obtained from Sinopeg (Xiamen, China).

### 2.2. Biological Reagents

Triton X-100 and Cell Counting Kit-8 were purchased from Beyotime Biotechnology (Shanghai, China). RiboGreen, Eagle’s minimum essential medium (DMEM) and fetal bovine serum (FBS) were purchased from Sangon Biotech (Shanghai, China).

### 2.3. Synthesis of Disulfide Lipids 

First step: To a round bottom flask, 1,2-di(pyridin-2-yl)disulfane (PySSPy) (3.0 equiv) was dissolved in MeOH (0.5 M); then, 2,3-Dimercapto-1-propanol solution (1.0 equiv, MeOH 2.0 M) was added over 10 min. Stirred at room temperature for 12 h, the solution was removed and purified by flash chromatography on silica gel (DCM → DCM/EtOAc 10:1), yielding the desired product **4**. Second step: Mercaptan (3.0 equiv) was dissolved in DCM (0.1 M); then, **4** (1.0 equiv, DCM 2.0 M) was added. This was stirred at room temperature for 12 h; then, the solution was removed and purified by column chromatography on silica gel (EtOAc/DCM/n-Hexane 1:1:60 → EtOAc/DCM/n-Hexane 1:1:10), yielding the desired product **5**. Third step: **5** (1.0 equiv) was dissolved in DCM (0.05 M); then, DIPEA (1.5 equiv), DMAP (0.1 equiv) and amino acid (1.2 equiv) were successively added under 0 °C. This was stirred at 0 °C for 10 min, then, EDCI (1.8 equiv) was added and stirred at room temperature for 12 h. The solution was removed and purified by flash column chromatography on silica gel (DCM/MeOH 15:1), yielding the desired final lipid. 

### 2.4. Lipid Nanoparticle (LNP) Formulation

LNPs were prepared by using a microfluidic mixture (Inano E, Micro&Nano, Shanghai, China), as previously reported [26]. Briefly, one volume of lipid mixtures (ionizable lipid, DSPC, cholesterol, DMG−PEG at 50:10:38.5:1.5-mole ratio) in ethanol and three volumes of mRNA solutions (acetate buffer, pH = 4.6) were mixed through the micromixer at a combined flow rate of 12 mL/min. The resulting mixtures were diluted with four volume of PBS buffer and then concentrated by ultrafiltration. These two procedures were repeated three times. Finally, the concentration of the LNPs should be condensed to 4.0 × 10^−6^ M, in terms of all four lipids components.

### 2.5. Characterization of LNPs

The size and zeta potential of LNPs were determined by a Malvern Nano ZS Zetasizer (Malvern Instruments Ltd., Worcestershire, UK). Size and zeta potential measurements were operated in PBS (pH = 7.4). Encapsulation efficiency and concentration were measured by RiboGreen.

### 2.6. Cell Culture

Cells were maintained in DMEM, containing 10% (*v*/*v*) fetal bovine serum at 37 °C in a 5% CO_2_.

### 2.7. Firefly Luciferase Activity Assay

293T cells were seeded in 24-well plates at a density of 50,000 cells per well per 800 µL left to adhere overnight, respectively. Then LNPs, including 1.0 µg Fluc−mRNA, was added to the wells, respectively. The same amount of mRNA packaged with commercial transfection reagent Lipofectamine 2000 (Lipo2k), according to the instructions, was used as a positive control. One day later, the cells and culture solutions in the per well were collected and centrifuged at 13,000 r for 3.0 min. The liquid supernatants were removed and 100 µL cell lysis buffer was added. After 5 min, the samples were centrifuged at 12,000 r for 5 min. 20 ul of supernatant were transferred to a 96-well plate, followed by adding 20 µL detecting reagent. Finally, the luminescence was determined by the Luminometer. 

### 2.8. GSH-Triggered Bioreducible Lipid Degradation

The prepared LNP 20 ul (8.0 × 10^−6^ M, in terms of all four components) was diluted with 1 ml GSH, with the GSH concentration ranging from 1.0 × 10^−3^ M to 32 × 10^−3^ M. It was then incubated at 37 °C for 4.0 h, followed by the measurement of size and PDI. At the same time, the entrapment efficiency of all the LNPs were measured by using RiboGreen.

### 2.9. Measurement of pKa via TNS

The pKa of C4S18A derived LNP was determined using TNS. TNS solution was prepared in distilled water at a concentration of 100 μM. LNP were diluted to 100 μM lipid (for four components) in 1.0 mL of buffer, with the pH ranged from 2.5 to 11, containing 10 mM HEPES, 10 mM MES, 10 mM ammonium acetate, 130 mM NaCl. A few drops of the TNS solution was added to the buffer, giving a final concentration of 1 µM, followed by the determination of fluorescence intensity at room temperature via a microplate reader (BioTek, Synergy LX, Montpelier, VT, USA), using excitation and emission wavelengths of 321 nm and 445 nm. A sigmoidal best fit analysis in the software Origin was applied to the fluorescence data and the pKa was confirmed, according to the pH giving rise to the half-maximal fluorescence intensity.

### 2.10. Cell Viability

Cell Counting Kit-8 assays were used to quantify the influence of LNP on cell progression. Briefly, HepG2 cells or HEK 293T cells (12,000 per well) in DMEM medium (90 μL) were seeded overnight in 96-well plates and then incubated with different nanocarriers at the concentration of 16 × 10^−6^ M and 32 × 10^−6^ M (for four components). After 24 h incubation, CCK8 (10 μL) was added to the medium and incubated for 1 or 2 h, followed by the recordings of absorbance at 450 nm using a microplate reader (BioTek, Synergy LX).

### 2.11. In Vivo Bioluminescent Imaging

All experiments were conducted using female BALB/c mice (6–8 weeks old) from Guangdong Yaokang Biotechnology Co., Ltd. (Guangzhou, China). All animal experiments were carried out in line with the regulations approved by the Institutional Animal Care and Use Committee of the Southern University of Science and Technology (SUSTech–SL2021081302, 13 August 2021). For in vivo bioluminescence imaging, mice received a single intravenous (IV) or intramuscular (IM) injection of LNP at a firefly luciferase (Fluc) mRNA dose of 0.5 mg·kg^−1^. After 6 h, 12 h, 24 h, 48 h, d-luciferin potassium salt (150 mg·kg^−1^) were injected into mice intraperitoneally. Ten minutes later, mice were anesthetized by isoflurane and placed in the imaging chamber for whole-body imaging using an animal imaging system (PerkinElmer, IVIS Spectrum, Waltham, MA, USA).

## 3. Results

### 3.1. Design and Syntheses of Lipids

Our aim was to develop effective ionizable cationic lipids with new structures for mRNA delivery in vivo that could take effect in an efficient and safe manner. Normally, an ionizable lipid embodies three parts, the hydrophilic head, the hydrophobic tails and the joint group between them. Many chemical groups have been integrated into the lipid as a linkage, such as acetal, ester, amide and so on [17,18,27]. The disulfides are naturally occurring and have good biocompatibility. Therefore, it is promising to introduce disulfide bonds into lipids. Starting from the commercially available dimercaprol, we established a library of disulfide bond-containing lipids by employing a series of thiols with variable carbon chains as hydrophobic tails and different kinds of amines as heads. The pivotal intermediate **2** (Figure 2a) was obtained via the activation of the sulfhydryl group in the presence of 2,2’-di-thiodipyridine (PySSPy) [28]. With enough common intermediate **2** in hand, it was attacked by an array of nucleophilic thiols (10, 12, 14, 16, 18 carbons alkyl chains) and the disulfide bond was formed to give **3**; then, the desired disulfide lipid **1** was achieved through an esterification reaction with the aid of EDCI as a coupling reagent. All the lipids were prepared by employing these standard synthetic procedures. The hydrophilic heads and hydrophobic tails of the new lipids were shown in Figure 2b,c. The tails are termed as S and the numbers behind them are referred to the number of carbons. “C” stands for the ionizable head part of the lipid, the figure behind ‘’C’’ meaning the number of carbons between the ester bond and the nearest nitrogen bond. The structures of the synthesized lipids were confirmed by NMR and high-resolution mass spectroscopy.

### 3.2. Screening of LNPs

After obtaining enough ionizable cationic lipids, we started to prepare the LNPs. The preliminary screening of the lipids was processed via the preparation of empty LNPs and the measurement of the physicochemical properties and stability. The four components (the disulfide ionizable lipid, cholesterol, 1,2-distearoyl-sn-glycero-3-phosphocholine and PEGylated lipids) were dissolved in ethanol in the molar ratios of 50:38.5:10:1.5, respectively, then, LNPs were prepared by using a microfluidic device as previously demonstrated [27]. The size of the LNPs was in the range of 50~130 nm (Table 1). LNPs obtained from saturated alkyl chain tails were liable to decompose and the storage life was less than 3 days at 4 °C (Entry 1–7), which is unsuitable for mRNA delivery. Linoleoyl tail is widely exploited in the development of excellent lipids, such as DLin-MC3-DMA, in which there are two double bonds [26]. The degree of unsaturation has been shown to influence mRNA delivery and stability. Next, we tried to use the unsaturated linoleic acid-derived lipid C4S18A (Entry 8) and oleic acid-derived lipid C4S18B (Entry 9) to produce LNPs, and they both showed better stability in comparison with the alkyl counterpart C4S18. C4S18A formulation shows evenly distributed, as identified by polydispersity index (PDI < 0.2); then, using the optimal S18A tail, we screened different hydrophilic heads to optimize the lipids. These LNPs (Entry 10–16) are stable for more than 3 days at 4 °C. Subsequently, five homogeneous LNP formulations (Entry 8, 10, 11, 14 and 16) were singled out to package the firefly luciferase (Fluc) mRNA with the N/P ratio of 7:1. The size of the resulting LNPs fluctuated between 80 nm and 120 nm, and PDI was usually less than 0.2 (Figure 3a), along with the zeta potential ranging from −1.5 to −6.0 mV (Figure 3b), which meets the requirement for the nano-drugs. Furthermore, the encapsulation efficiency (EE) was tested in the presence of Ribogreen, with all the EE over 85% (Figure 3c). For example, C4S18A-derived LNPs has a size of 115.0 nm and PDI of 0.133 (Figure 3d). A direct illustration of LNPs size of C4S18A was displayed by cryo-electron microscopy in Figure 3e. Next, we tried to confirm the Fluc-mRNA expression efficacy by incubating the mRNA-LNP in 293T cells in a 24-well plate. C3S18A, C3BS18A and C3DS18A gave lower luminance intensity compared with the commercial transfection reagent lipofectamine 2000 (Lipo), while the intensity of C4S18A and C4AS18A is higher than Lipo (Figure 3f). Subsequently, we prepared LNPs with different N/P ratios using C4S18A (Appendix A) and the highest luciferase protein expression (Figure 3g) was detected at an N/P ratio of 7/1. As shown in Figure 3h, the treatment of LNPs prepared by C4S18A with GSH in different concentrations ranging from 1.0 × 10^−3^ M to 32 × 10^−3^ M results in EE change, implicating different degree of mRNA release. No significant change of EE occurs when the GSH concentration is below 4.0 × 10^−3^ M, referring to that there exists a balance between the degradation and self-assembly. The balance indirectly verifies the good stability of our new LNPs. The speed of the mRNA release is dramatically accelerated as the GSH concentration over 8.0 × 10^−3^ M, until the next balance is formed with the GSH concentration over 16 × 10^−3^ M. In a word, EE experiments reveal that the intracellular reductive environment can trigger our LNPs degradation and facilitate intracellular mRNA release.

### 3.3. Stability and Biological Evaluation of mRNA-LNPs

According to the above data, we can see LNPs originated from C4S18A characterized by appropriate physicochemical properties and relatively high expression efficiency. Thus, LNPs from C4S18A were determined for more tests. Firstly, the stability of these two LNPs was explored by recording the size, PDI and EE over time when storing the samples at 4 °C. No significant change in size and EE was observed (Figure 4a,b). Although the PDI rises slightly over time, it keeps below 0.2, even after 40 days (Figure 4c). Next, we measured the apparent pKa of C4S18A LNPs using the fluorescence intensity of 2-(p-toluidino)-6-naphthalene sulfonic acid (TNS). The pKa of LNP is an important parameter for the cationic lipid and most of the efficient cationic lipids are characterized by apparent pKa values between 6 and 7 [29]. The pKa of C4S18A LNPs was determined to be 6.17 by a curve fit (Figure 4d). Moreover, we surveyed the toxicity of LNPs to cells and cell counting kit−8 (CCK8) assays were performed in 293T and HepG2 cells (Figure 4e). Cells were incubated with LNPs at a high concentration of 16 and 32 uM. We did not observe any toxicity, and cell viability was always more than 90%, compared to the PBS blank control, which confirmed the good biocompatibility and safety of our lipids.

### 3.4. In Vivo Experiments

Having confirmed the LNPs’ good stability and in vitro transfection ability, we resorted to evaluating delivery efficacy in vivo. BALB/c mice were injected with a single dose of LNPs containing 0.5 mg/kg of Fluc mRNA via intravenous injection (IV) or intramuscular injection (IM). After 6 h, 12 h, 24 h, 48 h, the luminance intensity was monitored, respectively, via an In Vivo Imaging System (IVIS). Strong signals were observed at 6 h for both C4S18A and C4AS18A LNPs, which waned gradually over time (Figure 5a–c). The majority of the signal was observed in the liver for IV injection, and at the leg for IM administration. The luminance intensity almost disappeared after 48 h, demonstrating the good metabolic capacity of LNPs. For both IV and IM injections, C4S18A showed stronger delivery ability than C4AS18A. Mice were also sacrificed 6 h after IV injection to analyze the biodistribution of LNPs. Almost no signal was observed in the lung, heart and kidney, but the luminance was strong in the liver and moderate in the spleen (Figure 5d). Taking the animal experiments together, our LNPs show excellent delivery and translation capability without harming the mice, as LNPs can be metabolized and cleared by the blood circulation and liver quickly.

## 4. Conclusions

In conclusion, a library of 16 ionizable lipids containing disulfide bonds is established via a 3-step synthetic procedure, which is cost-effective and time-saving. The preliminary screening is completed by monitoring the size, PDI and stability of LNPs. The hydrophobic tail, including the double bond, is superior to the saturated alkyl chain. The leading LNPs formulations C4S18A, which both embody linoleoyl tails, were identified by in vitro screening. The LNPs were stable at 4 °C for 40 days and they showed minimal cell cytotoxicity, which is verified by CCK8 assays using 293T cells and HepG2 cells. When the Fluc−mRNA-loaded LNPs prepared by C4S18A and C4AS18A were injected into mice, our samples present excellent luminance intensity. These discoveries offer an alternative to developing biodegradable and highly efficient nanocarriers for mRNA delivery. Further study on the structure–functionality relationships of this kind of LNP is currently underway.

## Figures and Tables

**Figure 1 pharmaceutics-15-00477-f001:**
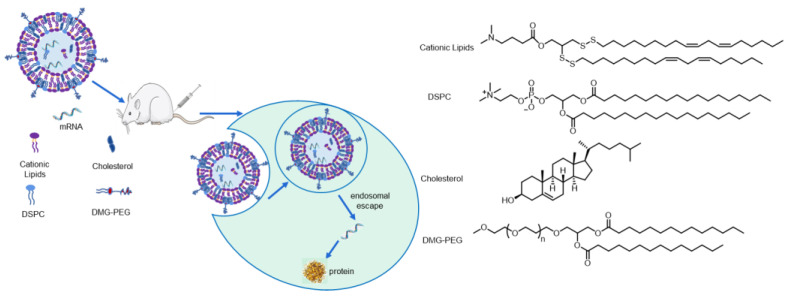
Schematic illustration of LNPs with disulfide bond-containing cationic lipids for mRNA delivery.

**Figure 2 pharmaceutics-15-00477-f002:**
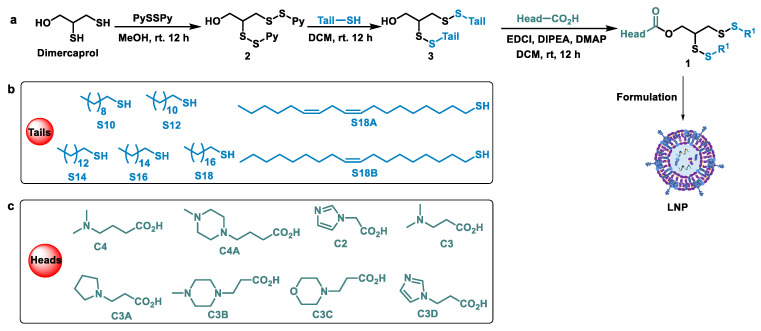
Construction of disulfide lipids library. (**a**) Synthetic route for the preparation of disulfide lipids. (**b**) Hydrophobic tails used in the lipids. (**c**) Hydrophilic heads used in the lipids. The heads and the tails were termed as C and S, and the numbers behind them refer to the number of carbons.

**Figure 3 pharmaceutics-15-00477-f003:**
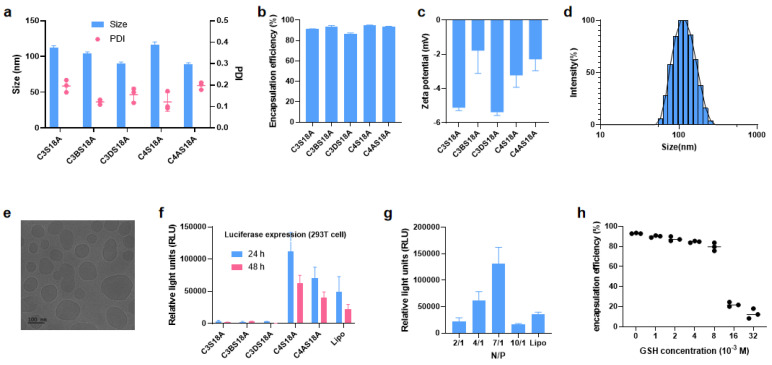
Optimization and characterization of LNPs prepared from leading lipids loaded with Fluc−mRNA. (**a**) Size (nm) and PDI of Luc−mRNA LNPs. (**b**) Zeta potential of Luc−mRNA LNPs. (**c**) Encapsulation efficiency (EE) of different LNPs loaded with Fluc−mRNA. (**d**) Size distribution by the intensity of C4S18A-derived LNP. (**e**) Cryo-electron microscopy of C4S18A LNP. (**f**) Luciferase expression following treatment of 293T cells with Fluc−mRNA LNPs. (**g**) Optimization of N/P ratio using the best C4S18A LNPs. (**h**) EE change over the concentration of GSH (from 0 to 32 × 10^−3^ M). Data presented as mean ± SEM of 3 separate samples or experiments.

**Figure 4 pharmaceutics-15-00477-f004:**
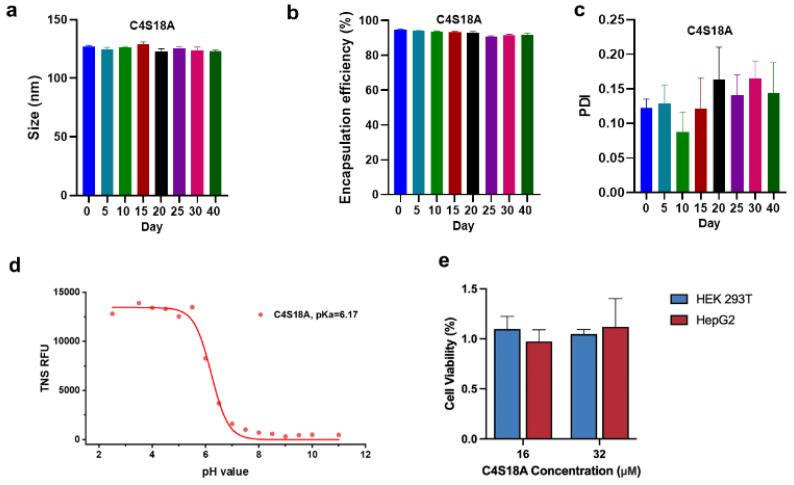
Stability and biological evaluation of mRNA-LNPs. (**a**) Particle size, (**b**) Entrapment efficiency (EE) and (**c**) PDI of LNPs prepared by C4S18A change over time. (**d**) The apparent pKa of LNPs was measured by plotting the TNS fluorescence against pH values. (**e**) Cell viability measured by CCK8 assays using 293T cells and HepG2 cells. Data presented as mean ± SEM of 3 separate experiments.

**Figure 5 pharmaceutics-15-00477-f005:**
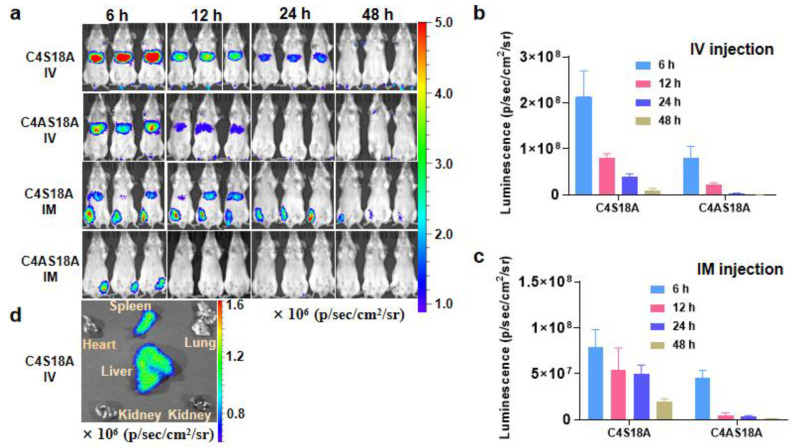
In vivo assays of C4A18A and C4S18A via intravenous and intramuscular administration. (**a**) Bioluminescence imaging of B/c mice (n = 3) over time after a single intravenous (IV) or intramuscular (IM) injection of LNPs at 0.5 mg kg^−1^ firefly luciferase mRNA dose. (**b**,**c**) Quantification of luminance intensity of Fluc−mRNA LNP over time via IV and IM administration. (**d**) Luminance intensity distribution among different organs 6 h after IV injection. Photon flux was quantified from ROI analysis. The data are representative of at least three independent experiments, and error bars indicate the SEM.

**Table 1 pharmaceutics-15-00477-t001:** The characterization of empty LNPs.

Entry	Name	size (nm)	PDI	Stability	Entry	Name	Size (nm)	PDI	Stability
1	C4S10	100 ± 23.1	0.266	II	9	C4S18B	62.1 ± 1.29	0.338	I
2	C4S12	52.0 ± 0.49	0.251	II	10	C4AS18A	104 ± 1.88	0.131	I
3	C4S14	102 ± 3.79	0.139	II	11	C3S18A	101 ± 1.41	0.207	I
4	C4S16	68.1 ± 0.62	0.120	II	12	C2S18A	63.2 ± 0.85	0.323	I
5	C4S18	122 ± 2.05	0.128	II	13	C3AS18A	101 ± 1.41	0.267	I
6	C3S14	89.6 ± 2.84	0.297	II	14	C3BS18A	100 ± 1.99	0.183	I
7	C3S16	110 ± 2.81	0.383	II	15	C3CS18A	82.8 ± 1.26	0.325	I
8	C4S18A	103 ± 1.03	0.177	I	16	C3DS18A	64.7 ± 1.08	0.146	I

I: Stable for > 3 day at 4 °C; II: Stable for < 3 day at 4 °C.

## Data Availability

Data are contained within the communication.

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
