# Peer review of "Development of a Library of Disulfide Bond-Containing Cationic Lipids for mRNA Delivery"

_pharmaceutics, 2023, doi:10.3390/pharmaceutics15020477_

Round 1

Reviewer 1 Report

Zhigao Shen et al. reported on the development of a library of disulfide bond-containing cationic lipids for preparing mRNA-LNPs. They developed a series of new ionizable lipids containing bioreducible disulfide bonds and constructed a library of lipids. Their LNPs that was composed of the ionizable cationic lipids could be stored at 4oC. They showed the importance of unsaturated fatty acid scaffold for the particle formation and in vivo gene delivery. This manuscript was well organized and concise. However, there are several points need to be addressed.

Important points

1) In my opinion, the most important characteristic of disulfide bond containing LNPs is their responsiveness to the reducing environment. So, I think the authors should show the change in physicochemical properties when GSH or DTT is applied to their particles, at least for the LNPs composed of C4S18A.

2) In the case of mRNA-LNP, unlike siRNA-LNP, etc., gene transfection efficiency often decreases even if there is no change in size, encapsulation rate, or PdI. This is considered to be partly due to the deterioration of mRNA itself during storage. The authors showed that the physicochemical properties of the particles did not change for 40 days. Do the authors have some data concerning to the stability of gene expression activity cells such as 293T?

Minor points

3) In line 80, the authors said “The heads and the tails were termed as C and S and the numbers behind them are referred to the number of carbons.”. I could understand that “S + number” means the length of the hydrocarbon chain of the thiol tails. However, the naming of “C + number” was a little confusing. Could the authors add some more explanation about the naming of the head group? What is C?

4) Line 95-97, the authors said “LNPs obtained from saturated alkyl chain tails were liable to decompose and the storage life was less than 3 days at 4 oC (Entry 1-7)”. The relationship between the extent of saturation of disulfide containing lipids and stability of LNPs is very interesting. Does the author have any idea about the reason? Also, please specify what index you used for evaluating the decomposition of empty particles during storage. (size/PdI or appearance of aggregation?)

5) In Figure 3g, why did the gene expression peak at an N/P ratio of 7/1 and then drop sharply at an N/P ratio of 10/1? Does the author have any idea about the reason?

6) Did the authors use ultrafiltration or tangential flow filtration to replace the external buffer? I think this information should be included in the methods section.

Reviewer 2 Report

The paper is well written and as written by Authors, future studies will provide more strucure-function relationships.

Comments:

0.5 mg/kg seems a high dose of mRNA, particularly for IM injection.

Stability of LNPs should be demonstrated by transfection after storage.

Lipofectamine Messenger Max is a more relevant standard for mRNA transfection, lipofectamine 2000 is outdated.

Y axis is hard to read in Figs 5B/5C.

Authors should verify that all the LNPs tested in Fig 3f are capable of mRNA intracellular delivery before comparing their transfection efficiencies.
